# Expert-guided Clinical Text Augmentation via Query-Based Model Collaboration

## Abstract

Data augmentation is a widely used strategy to improve model robustness and generalization by enriching training datasets with synthetic examples. While large language models (LLMs) have demonstrated strong generative capabilities for this purpose, their applications in high-stakes domains like healthcare presents unique challenges due to the risk of generating clinically incorrect or misleading information. In this work, we propose a novel query-based model collaboration framework that integrates expert-level domain knowledge to guide the augmentation process to preserve critical medical information. Experiments on clinical prediction tasks demonstrate that our lightweight collaboration-based approach consistently outperforms existing LLM augmentation methods while improving safety through reduced factual errors. This framework addresses the gap between LLM augmentation potential and the safety requirements of specialized domains.

## 1 Introduction

Data augmentation is a promising approach for enhancing model robustness by expanding training datasets with synthetic examples. The augmented data is expected to preserve essential semantics while introducing task-irrelevant variations, enabling the model to focus on core task-relevant features, thus improving robustness and generalization across diverse contexts (Cheng et al., 2019; Chen et al., 2021). However, in expert-driven applications such as healthcare and law, the use of data augmentation presents unique challenges. These applications demand a high standard of consistency and safety, whereas hallucinated information in augmented data, such as fabricated patient symptoms or false vital signs, can confuse models and propagate errors that potentially impact critical decisions (Kim et al., 2025). Therefore, data augmentation must be carefully controlled and validated to maintain data integrity and prevent the introduction of misleading or harmful information.

Researchers have increasingly adopted LLMs for generating synthetic text data due to their concept-understanding and instruction-following capabilities (Dai et al., 2025; Feder et al., 2023; Li et al., 2024b; Si et al., 2025). The preference for LLM usage is also from inherent challenges of data augmentation in natural language processing tasks, where traditional static augmentation techniques, e.g., synonym substitution, are not broadly effective (Okimura et al., 2022). Despite their usefulness, LLM factual errors remain a persistent issue: Generated text may alter critical information in the original text or produce false content (Shen et al., 2023; Yu et al., 2023). While these risks are well-documented, existing methods for ensuring the safety and reliability of LLM-augmented data in high-stakes applications remain inadequate, lacking domain-specific safeguards.

In this paper, we examine the distinctive requirements for LLM-based data augmentation in high-stakes domains, with a focus on preserving critical information and ensuring factual correctness. Our study centers on clinical note processing for medical applications, where LLMs have been used to generate counterfactual notes to improve clinical prediction model training (Feder et al., 2023). However, general-purpose LLMs often lack the domain expertise necessary to produce safe, high-quality synthetic data. To address this challenge, we propose a novel data augmentation framework that achieves both safety and efficiency through model collaboration (Li et al., 2024a): we inject expert-level knowledge via a lightweight "weak expert" model (BERT-based) that supervises the LLM's generation process. This approach provides domain-specific safeguards for improved augmentation quality while maintaining computational efficiency. We empirically show that our proposed augmentation method using dual model-collaboration produces safer and factually consistent

augmented data, outperforming existing baselines across multiple benchmarks and tasks. Lastly, we show that our collaborative method (built from pre-trained models with no additional training) can be distilled into a single model via preference learning (Rafailov et al., 2024), offering a trainable alternative that broadens the applicability of our method across different deployment settings. We state our contribution as follows:

- We propose a novel model collaboration framework for safe clinical text augmentation, where LLM generation is guided by a lightweight domain expert model to preserve critical medical information.
- We demonstrate our method's effectiveness across multiple dimensions: quantitatively improving safety through reduced medical term deletion and fewer irrelevant term introductions, while outperforming existing LLM-based augmentation methods across multiple clinical tasks.
- We extend beyond inference-time collaboration by demonstrating that our framework supports preference-based reinforcement learning to elicit generalist models to function as expert models.

## 2 RELATED WORK

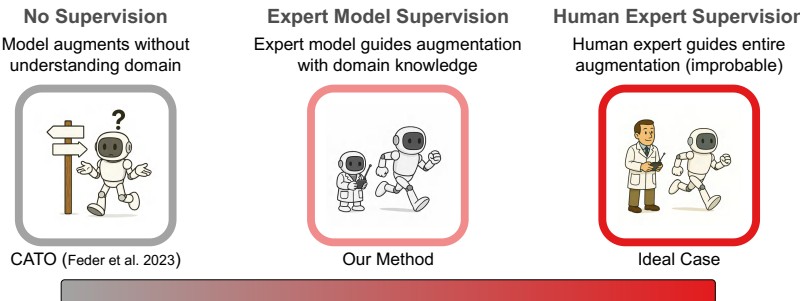

Figure 1: LLM-based Augmentation methods fail when data requires expert domain knowledge. Previous methods like CATO (Feder et al., 2022) perform data augmentation without supervision, resulting in errors such as keyword removal and factual mistakes due to lacking expertise. While human experts (e.g., caregivers) would be ideal supervisors, their limited availability and high cost make this impractical. We propose model collaboration as an intermediate solution: an *expert model* trained on domain data substitutes human experts, guiding augmentations by extracting domain knowledge from clinical text and injecting them into inference queries.

**Clinical Language Models**    Clinical language models have emerged as an important foundation for advancing natural language processing (NLP) applications in healthcare. Researchers have adapted transformer-based language models to process biomedical and clinical texts, including ClinicalBert (Huang et al., 2019), BioBert (Lee et al., 2020), GatorTron (Yang et al., 2022), and NYUTron (Jiang et al., 2023), by domain-specific pre-training on large-scale electronic health records (EHRs) and medical literature. These models can be fine-tuned with minimal architectural modification for downstream tasks and have demonstrated improved performance on a wide range of clinical tasks, e.g., hospital readmission prediction and medical named entity recognition. The models address conventional methods' reliance on structured EHR and the complexity in feature and algorithm development (Kelly et al., 2019), by interpreting useful clinical information from unstructured clinical notes for a variety of prediction tasks. Despite these advances, the robustness of clinical language models remains a challenge; models often struggle to generalize across different institutions, patient populations, and documentation styles (Moradi & Samwald, 2022; Rahman et al., 2024), which are critical to consider when developing models to inform real-world healthcare decisions. In response, we propose a data-centric approach to improve generalization and address distribution shifts for robust application of clinical language models in the real-world.

**Data Augmentation**    Data Augmentation is an effective technique to improve model robustness, where the key is to create diverse augmented versions of the original data while maintaining its semantic integrity (Geiping et al., 2022; Feng et al., 2021), whose difficulty varies by modality (e.g., image, text). For instance, image data benefits from its intrinsic spatial correlations and inherent redundancy, making it less vulnerable to feature distortions introduced during augmentation (Per-

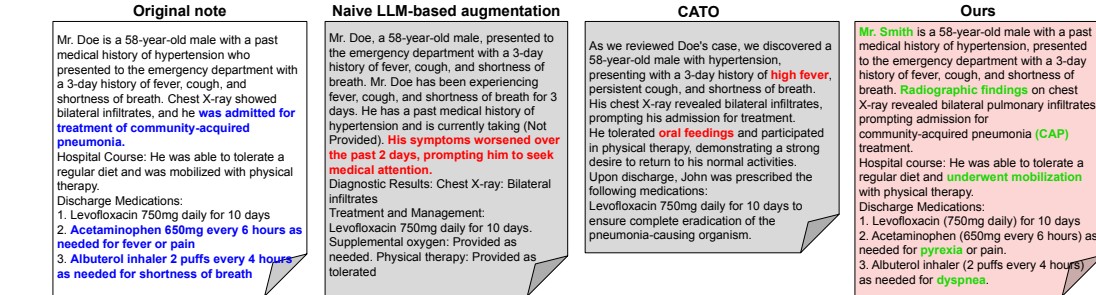

Figure 2: Synthetic clinical notes generated by different LLM-based augmentation methods are shown: a simple rephrasing prompt (Naive LLM-based augmentation), prompting to rewrite the note by only changing the physician's writing style (CATO), and our model-to-model query method (Ours). Notes generated by the Naive and CATO methods omit critical medical information (highlighted in blue) and introduce hallucinated and irrelevant content (highlighted in red). In contrast, our method preserves all medical information while only rephrasing non-critical elements (e.g., patient names, synonyms of medical terms; green), achieving safe clinical note augmentation. *Note:* The original note shown here is synthetic and not a real note from our used dataset.

vin et al., 2021; Cho et al., 2025). On the other hand, text data augmentation faces challenges in maintaining semantic integrity during augmentation (Chai et al., 2025; Dai et al., 2025), owing to its syntactic attributes (Chen et al., 2023) (e.g., grammar, context) which should not be perturbed, especially in safety-critical domains (e.g., healthcare (Nazi & Peng, 2024)). To address such short-comings, recent works focus on semantic-aware data augmentation that does not change the key components of the text, namely through simple semantic-preserving transformations (Van et al., 2021; Chen et al., 2023) (e.g., synonym replacement, random swapping), or model-based augmentation techniques that utilize large language models (LLMs) to produce fine-grained augmentations (Chai et al., 2025; Li et al., 2024b; Yoo et al., 2021; Zhou et al., 2021). Notably, Feder et al. (2023) presents a semantic-preserving augmentation method that incorporates LLMs to augment non-causal features (e.g., writing styles). However, the studies do not address common limitations of LLMs (e.g., hallucinations (Yao et al., 2024) and spurious correlations (Zhou et al., 2023)), which remains an issue in guaranteeing semantic-aware, *safe* augmentation. In this paper, we specifically study cases where the LLMs fail to differentiate between critical and non-critical information, tampering with the semantics of the original sample and endangering the safety of the model trained with distorted data.

## 3 PROBLEM FORMULATION

**Causally Driven Data Augmentation.** In many safety-critical applications (e.g., clinical or legal domains), we often have additional domain knowledge or causal assumptions that elucidate which variables (tokens, phrases) truly affect the label $y$ (Feder et al., 2023; Staliūnaitė et al., 2021). We can depict these dependencies with a causal graph $\mathcal{G}$ (as shown in Figure 3), either explicitly specified by experts or derived via observational data. Formally, we posit that the label $y$ depends on a set of domain variables $\mathcal{V}$ (e.g., symptoms, diagnoses for clinical data) and that altering any of these crucial variables could distort the semantics. Conversely, stylistic or non-critical variables $\mathcal{U}$ (e.g., function words, phrasing) do not affect $y$, although they may still correlate with it (e.g., due to shared confounders). As a result, models may rely on $\mathcal{U}$ as shortcut features which can lead to unreliable predictions. Data augmenta-

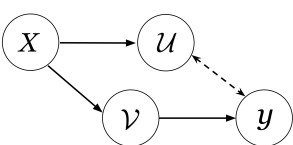

Figure 3: Clinical language model predictions ($y$) are influenced by both meaningful domain variables ($\mathcal{V}$) and spurious features ($\mathcal{U}$) extracted from note data ($X$).

tion approaches have been aiming to generate counterfactual examples by altering $\mathcal{U}$ to decorrelate it from $y$, thereby encouraging the model to make predictions unaffected by non-critical information.

**Pitfalls of LLM-based Augmentation Methods** However, an under-explored challenge in using LLM-based methods to generate counterfactual examples (Feder et al., 2023; Zhou et al., 2024)

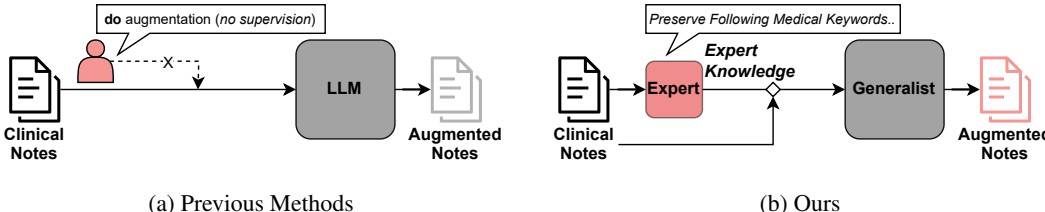

(a) Previous Methods                                (b) Ours

Figure 4: Comparison of augmentation strategies. (a) Previous methods do not provide supervision over the augmentation process, assuming the LLM has expert-level knowledge. (b) Our augmentation method leverages query-based model collaborations to provide domain knowledge of the weak expert model to guide the strong generalist model within an LLM-based augmentation module.

lies in the inability of general-purpose LLM to precisely distinguish between observed variables $\mathcal{V}$ and unobserved variables $\mathcal{U}$ in the data. Therefore, there is a growing disconnect between the theoretical frameworks for robust learning and the practical implementation of augmentation. Unlike image augmentation which commonly uses determined algorithms, text augmentation using LLMs introduces variability and hallucination to generated text which may undermine the robustness assumptions of models trained with the augmented data. For example, LLMs often lack the domain-specific understanding on medical language to reliably preserve critical clinical information while only modifying/augmenting non-medical parts in clinical notes (failure examples in Figure 2). This limitation causes LLM-based augmentation methods to lose intended causal control and may introduce semantic distortions (Ding et al., 2024; Sriramanan et al., 2024; Song et al., 2024). While issues of error and hallucination in LLMs have been discussed (Tonmoy et al., 2024), little work has addressed their impact on the safety of data augmentation. We fill this gap by introducing explicit guidance for LLM inference during augmentation through a collaborative framework, thereby reducing hallucinations in causally-informed data augmentation.

## 4 MODEL-TO-MODEL QUERY FOR FINE-GRAINED DATA AUGMENTATION

In this section, we present our model-to-model query framework for LLM-based textual data augmentation. We begin by introducing the notation and two core components (*weak expert* and *strong generalist*). We then describe how their outputs are integrated into a unified augmentation pipeline, and conclude by discussing why this design enables safer and more domain-targeted augmentations compared to existing approaches.

### 4.1 NOTATION AND SETUP

Let $\mathcal{D} = \{(x_i, y_i)\}_{i=1}^N$ be a dataset of $N$ labeled text samples, where each input $x_i$ is a raw text (e.g., a sentence or document) and $y_i$ is its annotation or label. Our goal is to construct an augmented dataset $\tilde{\mathcal{D}} = \{(\tilde{x}_i, y_i)\}_{i=1}^N$, where each $\tilde{x}_i$ preserves the semantic of $x_i$, particularly its *critical* domain tokens, et modifies its non-critical tokens, e.g., surface style or phrasing. The distinction between critical and non-critical tokens is informed by a prior causal graph grounded in domain literature for the specific clinical task (Figure 3). $X$ denotes the original clinical notes which includes both predictive factors relevant to the task $\mathcal{V}$ and spurious factors $\mathcal{U}$ that typically do not generalize. In our setup, we only specify $\mathcal{V}$ since only these tokens are preserved during LLM-based augmentation. We define $\mathcal{V}$ as medical-clinical terms, e.g. disease disorder and sign symptom, which are predictive to clinical prediction tasks indicated by literature (Davis et al., 2022; Gao et al., 2023)

By referencing causally driven augmentation model $\mathcal{G}$, we incorporate two components: a weak expert $W$ which identifies critical variables in the input $X$ and flags them as unalterable tokens, and a strong generalist $G$ with strong generative capability to write counterfactual clinical notes:

1. **Weak Expert** $W(\cdot)$: A lightweight domain-specific model (e.g., a BERT-based clinical language model) that identifies safety-critical tokens (i.e., medical keywords) which must remain unchanged during augmentation.
2. **Strong Generalist** $G(\cdot)$: A general purpose foundation model with strong rewriting and generative capabilities but without explicit training in the target domain.

We treat the weak expert as a domain-sensitive decision-maker that constrains critical content, and the strong generalist as a general-purpose rewriter guided by these constraints.

### 4.2 Formalizing the Framework

To generate an augmented text $\tilde{x}_i$ from a input text $x_i$, our pipeline consists of three steps:

**A. Critical Features Extraction by Weak Expert.** The weak expert $W$ identifies the key tokens or features in $x_i$ that are essential for preserving semantic fidelity:

$$\mathcal{K}_i = W(x_i). \tag{1}$$

The set $\mathcal{K}_i$ typically include terminology or clinical expressions that *must not* be altered in order to maintain the original meaning.

**B. Prompt Construction.** We create a prompt using the template shown on the right: $\text{prompt}(x_i, \mathcal{K}_i)$ that passes the original text and explicit constraints provided by the weak expert $W$ to the strong generalist $G$. Formally, the prompt specifies the set $\mathcal{K}_i$, highlighting the domain-critical terms whose meanings must remain unchanged.

> **Original text:** $x_i$
>
> **Preserve the following tokens:** $\{\mathcal{K}_i^1, \mathcal{K}_i^2, \cdots\}$
>
> **Rewrite instructions:** Rewrite $x_i$ in a new style or phrasing *without* altering any token in $\mathcal{K}_i$.

**C. Safer Text Rewriting by Strong Generalist.** The strong generalist $G$ generates the rewritten text $\tilde{x}_i$ by conditioning on the constructed prompt:

$$\tilde{x}_i = G\Big(\text{prompt}(x_i, \mathcal{K}_i)\Big). \tag{2}$$

We pair $\tilde{x}_i$ with the original label $y_i$ to form the augmented dataset $\tilde{\mathcal{D}} = \{(\tilde{x}_i, y_i)\}_{i=1}^N$. Because $G$ receives explicit guidance on domain-critical tokens, t avoids distorting key information while freely rephrasing non-critical content. In this way, a small, specialized model $W$ contributes domain knowledge and safety constraints, while the strong generalist $G$ executes the generative rewriting. In Appendix A.1, we report the details of our method implementation (e.g., the user prompts)

## 5 Experiments

### 5.1 Datasets and Benchmarks

We use the MIMIC-III dataset (Johnson et al., 2016), a widely used public resource of de-identified clinical notes. This dataset provides a diverse collection of clinical documentation including discharge summaries, nursing notes, and physician reports, making it an ideal testbed for evaluating augmentation techniques for clinical text. We consider three clinical prediction tasks: (1) 30-day all-cause readmission prediction, estimating the likelihood of patient returning to hospital within 30 days following discharge. This task is both clinically and operationally significant (Caruana et al., 2015; Kansagara et al., 2011), reflecting how well language models capture meaningful representations from clinical notes (Huang et al., 2019). (2) In-hospital mortality prediction, predicting all-cause death during hospitalization, useful for disease management (Ke et al., 2022). (3) Hospital length-of-stay prediction, predicting the number of days a patient will remain in hospital during a single admission event, a major indicator for the consumption of hospital resources (Stone et al., 2022).

Besides training downstream prediction models using augmented data, we also evaluate in zero-shot inference settings. Specifically, (1) patient phenotyping, using phenotype annotations from Gehrmann et al. (2018), and (2) ICD clinical coding, where we follow prior work (Mullenbach et al., 2018; Zhang et al., 2025) to construct datasets (MIMIC-III-Full and MIMIC-III-Top-50).

### 5.2 Implementation

We use the biomedical-ner-all model (Raza et al., 2022) as the Weak Expert $W(\cdot)$. The model is built on DistilBERT architecture and trained to recognize 107 biomedical entities in clinical texts.

For the Strong Generalist $G(\cdot)$, we experiment with different instruction-tuned models (e.g., Qwen-3-0.6B (Yang et al., 2025) and LlaMA-3.2-3B-Instruct (Grattafiori et al., 2024)), which excel at rephrasing, summarizing, or restructuring text in a human-like way. Unless explicitly stated, Qwen-3-0.6B model is used as default $G(\cdot)$. To address long input lengths in MIMIC-III notes, we implement Cache-Augmented Generation (CAG) (Chan et al., 2025) to expand the context window of the generalist models, allowing the model to maintain coherence and preserve critical clinical information throughout the augmentation process. To assess the effectiveness of different augmentation strategies, we conduct downstream clinical prediction tasks using the augmented datasets. Specifically, we fine-tune a Qwen-3 model with LoRA adapters (Hu et al., 2021) and a BERT model (Jiang et al., 2023) with full fine-tuning. Lastly, the hyperparameters were selected using a grid search. In Appendix A.2, we provide a detailed analysis of the hyperparameters (see Table 6, Table 7, and Table 8).

## 5.3 EVALUATION METRICS AND BASELINES

In our experiments, we evaluate the proposed method along two dimensions: (1) the quality of the synthetic data generated by augmentation, and (2) the utility of the augmented data for downstream clinical tasks, assessed through model training and zero-shot/few-shot inference. The corresponding evaluation metrics are as follows.

**Quality of the Synthetic Data**

- Preservation Rate (PR) (i.e., how many medical entities are preserved during augmentation. Higher is better), where $\mathcal{E}_{\text{orig}}$ is the set of medical entities in the original data, $\mathcal{E}_{\text{aug}}$ is the set of medical entities in the synthetic data (Liu et al., 2024).
- Hallucination Rate (HR) (i.e., how many irrelevant medical entities not existing in the original data are generated. Lower is better.) (Liu et al., 2024)

$$\text{PR} = \frac{|\mathcal{E}_{\text{aug}} \cap \mathcal{E}_{\text{orig}}|}{|\mathcal{E}_{\text{orig}}|}, \quad \text{HR} = \frac{|\mathcal{E}_{\text{aug}} \setminus \mathcal{E}_{\text{orig}}|}{|\mathcal{E}_{\text{orig}}|}. \quad (3)$$

**Utility of Synthetic Data for Downstream Clinical Tasks**

- Clinical outcome prediction: Accuracy on 30-day all-cause readmission and in-hospital mortality prediction, when models are trained on synthetic data generated by different augmentation methods.
- Length-of-stay prediction: Root Mean Squared Error (RMSE) for hospital length-of-stay prediction under the same setting.
- Patient phenotyping: Zero-shot and few-shot prediction using synthetic clinical notes.
- ICD coding: Zero/one/few-shot prediction of ICD codes, formulated as an information retrieval task following the practice of Boyle et al. (2023).

**Baselines** The most relevant comparison baselines are LLM-based textual data augmentation methods. We compare our approach with (1) a naive augmentation strategy, which prompts the LLM to rephrase the original note without introducing substantive variation (denoted as "Naive"), and (2) a causally driven augmentation method ("CATO") that prompts the LLM to modify only writing style of notes (Feder et al., 2023).

## 5.4 EXPERIMENTAL RESULTS

In this section, we evaluate our model collaboration framework through comprehensive experiments. First, we validate that our method preserves domain-critical information during augmentation, demonstrating improved safety over unsupervised LLM approaches. Second, we show performance gains on downstream tasks, both when training with our augmented data and in zero/few-shot inference settings. Third, we analyze how different weak expert designs impact augmentation quality. Finally, we move beyond inference-time collaboration and show our framework is trainable, by distilling the expert guidance into a single model via preference learning. Together, these experiments demonstrate that expert-guided augmentation achieves both safety and effectiveness in clinical applications.

**Safety Validation: Preserving Critical Medical Information.** We provide a detailed investigation into the quality of synthetic data generated by our augmentation method, focusing on how it

preserves the critical medical information while preventing groundless information from being added. Specifically, we compare the PR (preservation rate) and HR (hallucination rate) of synthetic notes. As observed in Table 1, LLM-based augmentation methods in general tend to remove or add critical keywords (i.e., named entities) during augmentation. Figure 2 shows an example of this problem. The naive LLM-based augmentation method removes up to $51\%$ of task-critical keywords (medical keywords), while adding up to $75\%$ groundless keywords that do not appear in the original text.

In contrast, our proposed augmentation method is most effective in preserving relevant medical keywords while preventing the introduction of fabricated keywords (Table 1). To assess robustness, we test with LLMs of different sizes: Llama-3 model 1B and 3B (Grattafiori et al., 2024), thereby disentangling the effect of the LLM's inherent semantic understanding and generative capacity. As expected, the larger, and thus stronger LLM that performs augmentation achieves preservation rates (PR) and lower hallucination rates (HR). But across both settings, our method consistently outperforms the baselines, providing more accurate and safe data augmentation.

Table 1: Quality of synthetic notes generated by different augmentation methods, measured by entity preservation rate (PR) and hallucination rate (HR) across 300 samples.

| Method | LLama-3.2-1B | | LLama-3.2-3B | |
|---|---|---|---|---|
| | PR ↑ | HR ↓ | PR ↑ | HR ↓ |
| Naive | 0.48 | 0.75 | 0.51 | 0.59 |
| CATO | 0.47 | 0.77 | 0.62 | 0.72 |
| Ours | **0.66** | **0.43** | **0.79** | **0.33** |

Table 2: Downstream task performance (Acc./RMSE) of Qwen-3 and BERT model trained with augmented data using different methods. Bold indicates the augmentation method that provides the largest performance gain. We report the mean and standard error results across 5 runs.

| Model | Aug. Method | Readmission (Acc.) | Mortality (Acc.) | Length-of-stay (RMSE) |
|---|---|---|---|---|
| Qwen-3 | Zero-Shot | $0.511_{\pm 0.06}$ | $0.901_{\pm 0.03}$ | $73.277_{\pm 8.19}$ |
| | None | $0.526_{\pm 0.04}$ | $0.911_{\pm 0.04}$ | $17.835_{\pm 5.29}$ |
| | Naive | $0.520_{\pm 0.04}$ | $0.907_{\pm 0.04}$ | $16.357_{\pm 5.75}$ |
| | CATO | $0.552_{\pm 0.04}$ | $0.910_{\pm 0.03}$ | $18.677_{\pm 3.20}$ |
| | Ours | $\mathbf{0.599}_{\pm 0.03}$ | $\mathbf{0.917}_{\pm 0.02}$ | $\mathbf{15.563}_{\pm 3.26}$ |
| BERT | None | $0.721_{\pm 0.03}$ | $0.897_{\pm 0.04}$ | $15.403_{\pm 0.12}$ |
| | Naive | $0.736_{\pm 0.01}$ | $0.916_{\pm 0.01}$ | $13.572_{\pm 0.04}$ |
| | CATO | $0.730_{\pm 0.01}$ | $0.923_{\pm 0.003}$ | $13.504_{\pm 0.02}$ |
| | Ours | $\mathbf{0.757}_{\pm 0.01}$ | $\mathbf{0.929}_{\pm 0.03}$ | $\mathbf{13.110}_{\pm 0.06}$ |

**Performance Gains: Downstream Tasks and Zero/Few-Shot Learning.** Table 2 reports how different augmentation methods affect predictive performance across downstream clinical tasks. Our expert-guided augmentation (Ours) achieves the best mean performance across all three tasks. With Qwen-3, our method improves readmission accuracy to 0.599 (+0.047 over the strongest baseline, CATO), mortality accuracy to 0.917 (+0.006–0.007 over baselines), and reduces length-of-stay RMSE to 15.563. The improvements are consistent across model architectures: when switching from the decoder-only Qwen-3 to the encoder-only BERT, our approach continues to outperform baselines.

In contrast, Naive augmentation shows mixed benefits: it degrades performance on readmission and mortality prediction compared to no augmentation, while slightly improving length-of-stay RMSE. CATO similarly improves readmission accuracy but harms performance on mortality and length-of-stay prediction. These patterns suggest that unguided or heuristically guided augmentation can inject label-preserving but distribution (meaningful domain variables ($\mathcal{V}$) in Figure 3)-shifting noise that degrades generalization of the model. Incorporating expert knowledge yields clinically faithful augmentations that provide consistent and robust gains.

Beyond training downstream models with augmented data, we evaluate whether augmentations preserve information critical for inference in low-resource settings. Specifically, we assess zero and few-shot performance on phenotype classification and ICD coding. For phenotype classification, we compare F1 scores on the original samples ("None") and on augmented data (Figure 5). Naively augmented samples (i.e., unconstrained LLM paraphrasing) and CATO consistently reduce task scores, indicating that the critical information (e.g., medical keywords) is distorted in the augmented notes, making them less predictive for the phenotyping task.

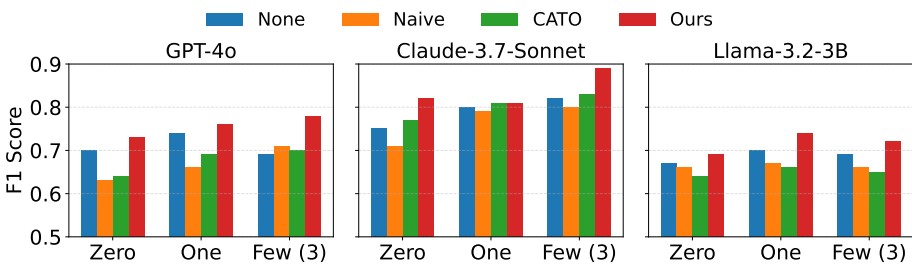

Figure 5: Zero/One/Few-shot F1 scores on Patient Phenotyping task using different inference models.

In contrast, our expert-guided augmentation reliably improves F1 across all zero/one/few-shot settings and inference models. For ICD coding (Table 3), we reframe the task into retrieval-based prediction, where the prediction is correct if the model can retrieve a grounding rationale for the ICD label from the text (Boyle et al., 2023). The observed pattern is similar: naive and CATO augmentations display lower scores than the original clinical notes (None in Table 3). But the aug-

Table 3: Recall (Rec.), Precision (Pred.) and F1 score on ICD Code Prediction. The task is framed as a retrieval task for zero-shot inference (Boyle et al. (2023)). We use GPT-4o as inference model.

| Aug. Method | Micro | | | Macro | | |
|---|---|---|---|---|---|---|
| | Rec. ↑ | Prec. ↑ | F1 ↑ | Rec. | Prec. | F1 |
| None | 0.221 | 0.159 | 0.185 | 0.178 | 0.197 | 0.187 |
| Naive | 0.146 | 0.138 | 0.149 | 0.133 | 0.146 | 0.139 |
| CATO | 0.153 | 0.141 | 0.147 | 0.166 | 0.173 | 0.169 |
| Ours | **0.224** | **0.168** | **0.192** | **0.189** | **0.203** | **0.196** |

mented samples produced by our method preserve performance: on par with or exceeding the original data. When our method performance exceeds the original data without using augmentation, this aligns with prior findings that LLM-rephrased texts can enhance predictive/learning signals by improving linguistic clarity (Deng et al., 2024; Pieler et al., 2024).

**Component Analysis: Effect of Weak Expert Design.**
We next examine how the weak expert model affects augmentation quality. In Table 4, we report the PR and HR score of synthetic samples generated with two types of expert models (1) medical-expert: a biomedical language model trained on domain data (Raza et al., 2022) and (2) general-expert: a general language model trained for named entity extraction. As expected, the medical-expert provides stronger guidance, leading to significantly higher preservation and fewer hallucinations. Another impressive observation is that even when the weak expert is a general language model without medical knowledge, our *expert*-collaboration framework still improves per-

Table 4: Effects of using different weak-expert models. Weak expert's proficiency in extracting expert knowledge affects the quality of data augmentation.

| Method | PR ↑ | HR ↓ |
|---|---|---|
| Naive | 0.51 | 0.59 |
| CATO | 0.62 | 0.77 |
| Ours (medical-expert) | **0.79** | **0.33** |
| Ours (general-expert) | 0.53 | 0.50 |

formance. This is likely because medical terms form a subset of named entities captured by the general-expert. This finding aligns with recent works related to weak supervision, where even imperfect learning signals are known to guide and benefit model training (Burns et al., 2023; Cho et al., 2025), highlighting the robustness and potential of our query-based collaboration framework.

**Framework Extension: Distilling Expert Guidance through RL.** Our central claim is that expert signals are the key driver of effective augmentation. So far, we have injected this signal at inference time through model collaboration (*weak expert W + strong generalist G*). To test whether this guidance can also be realized by a single model, we explore preference-based reinforcement learning (RL) as an alternative mechanism. Specifically, we train the generalist with direct preference optimization (DPO) (Rafailov et al., 2024), where the preference signal is defined to favor expert-guided over naive augmentations. The resulting model, denoted $W^*$, behaves as a *Strong Expert* that internalizes our augmentation method. Table 5 compares this RL-trained *Strong Expert* with our dual-model collaboration.

Table 5: Comparison of model-collaborative augmentation (Ours) against a single Strong Expert augmentation trained using reinforcement learning, on downstream task performance.

| Model | Aug. Method | Readmission (Acc.) | Mortality (Acc.) | Period (RMSE) |
|-------|-------------|--------------------|------------------|---------------|
| Qwen-3 | Zero-Shot | $0.511_{\pm 0.06}$ | $0.901_{\pm 0.03}$ | $73.277_{\pm 8.19}$ |
|  | Ours | $\mathbf{0.599}_{\pm 0.03}$ | $\mathbf{0.917}_{\pm 0.02}$ | $15.563_{\pm 3.26}$ |
|  | Ours (Strong Expert) | $0.582_{\pm 0.04}$ | $0.911_{\pm 0.03}$ | $\mathbf{15.482}_{\pm 4.17}$ |
| Llama-3.2-3B | Zero-Shot | $0.518_{\pm 0.05}$ | $\mathbf{0.904}_{\pm 0.02}$ | $80.839_{\pm 7.31}$ |
|  | Ours | $\mathbf{0.583}_{\pm 0.04}$ | $\mathbf{0.904}_{\pm 0.02}$ | $\mathbf{14.920}_{\pm 4.41}$ |
|  | Ours (Strong Expert) | $0.560_{\pm 0.06}$ | $0.901_{\pm 0.01}$ | $17.276_{\pm 4.68}$ |

We observe that preference learning *(Ours (Strong Expert) in Table 5)* improves over zero-shot baselines. However, the model collaboration (*Ours*) remains the most reliable overall across tasks and backbones. The gap between the single *Strong Expert* and the collaboration method is smaller for Qwen-3 than for Llama-3.2-3B. We hypothesize this is due to domain priors: Qwen models can already capture key medical terms from pretraining, so DPO-based preference learning better mimics *weak-expert + strong generalist* behavior. In contrast, Llama shows weaker keyword extraction, and RL-only training narrows but does not close the gap with the dual-model approach. When effective, preference learning compresses the dual-model policy into a single model that behaves like an expert augmenter. This shows that RL can elicit latent domain knowledge from a generalist and push its behavior toward expert-like augmentation (aligned with observations in Chu et al. (2025)). However, since the gains are inconsistent across backbones, we view a fully model-agnostic *Strong Expert* as an open question, and recommend the dual-model pipeline when base models lack medical priors.

## 6 Discussion: When does Model Collaboration Help?

We discuss *when* and *why* model collaboration with weak experts improves augmentation data quality. Weak experts $W$ are most helpful when augmentation must preserve specific domain terminologies while allowing flexibility in how the rest of the text is written. By identifying these critical tokens upfront, the generalist $G$ can vary style and phrasing without changing the medical meaning, achieving higher Preservation and lower Hallucination Rates (Table 1).

The benefits are particularly strong in these scenarios. First, in low-resource settings with rare conditions or drug–dose pairs appear uncommon in pretraining data, even a lightweight detector trained on medical text prevents deletion or ambiguous paraphrasing. Across weak-expert variants, medical specialization yields the best rresults, though general NER provides gains by identifying entity boundaries (Table 4). Second, under distribution shift across hospitals or time periods, weak experts preserve causal features while allowing style adaptation, improving downstream performance for prediction readmission, mortality and length of stay (Table 2). Third, in safety-critical applications, token-level guidance reduces hallucinations from paraphrase-based augmentation, as shown by stronger zero/one/few-shot phenotyping and improved ICD retrieval (Figure 5, Table 3).

Importantly, effectiveness depends on calibration. When weak experts under-detect, medical facts change; when they over-detect, augmentation variation is limited. In practice, the most reliable gains occur when the weak expert achieves high recall on safety-critical entities while preserving flexibility elsewhere. Under this balance, we see consistent improvements in augmented data quality and downstream tasks across backbones (Tables 1 and 2).

## 7 Concluding Remarks

In this paper, we introduce a query-based model collaboration framework that injects expert clinical knowledge into LLM data augmentation. By explicitly preserving domain-critical semantics while perturbing only task-irrelevant details, our approach produces safer, higher-quality synthetic notes. Experiments across diverse clinical tasks demonstrate consistent gains over standard LLM augmentation with markedly reduced hallucination and omission. These results show that coupling LLMs with lightweight expert guidance bridges the gap between LLM generative power and the strict accuracy requirements of high-stakes domains.

## ETHICS STATEMENT

The authors acknowledge and concur with the ICLR Code of Ethics, namely in its pursuit of (1) human well-being, (2) high standards of scientific excellence, (3) consideration for the societal impacts (i.e., harms) of AI, (4) honesty & trustworthiness, (5) fairness, (6) mutual respect for other researchers' works, (7) privacy, and (8) confidentiality.

## REPRODUCIBILITY STATEMENT

For reproducibility, we provide the source code, experimental guidelines, and the scripts used in our experiments. Please refer to the README.md file in the supplementary materials on how to reproduce our experiments. We also used a fixed seed setting, which is implemented in the source code. We also include notebook (.ipynb) files to reproduce the figures appearing in our paper. Lastly, in Section 5, we thoroughly explain how our method and its experiments are implemented.

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

# A APPENDIX

## A.1 METHOD DETAILS

> **System role:**
> You are a medical AI assistant with expertise in clinical documentation. Your task is to rewrite clinical notes while maintaining complete medical accuracy.
>
> **Important instructions:**
>
> - You must preserve all medical entities exactly as they appear.
> - Do *not* list or enumerate the entities — incorporate them naturally into the rewritten text.
> - You may change sentence structure, word choice, and writing style.
> - Do *not* change any medical terminology, dosages, measurements, or clinical findings.
> - Ensure the rewritten note contains the same medical information as the original.

> **Original clinical note:**
> {note}
>
> **Medical entities to preserve (verbatim):**
> {extracted_keywords}
>
> **Rewrite instructions:**
> Rewrite the original clinical note while *naturally* incorporating all listed medical entities. Do not list the entities separately. Maintain complete medical accuracy and do not alter any medical terminology, dosages, measurements, or clinical findings. Ensure the rewritten note conveys the same medical information as the original.

In this section, we provide the implementation details of our method. Our augmentation method instantiates the model collaboration framework defined in Section 4. A domain-focused weak expert $W$ first extracts safety-critical clinical entities from the input note $x_i$, producing a constraint set $\mathcal{K}_i = W(x_i)$. These entities (diagnoses, symptoms, medications, measurements) are treated as unalterable (i.e., should be preserved) during rewriting. We then construct a constraint-aware prompt that includes the original note and an explicit instruction to preserve every token in $\mathcal{K}_i$ verbatim. A strong generalist $G$ receives this prompt and generates $\tilde{x}_i$, which is paired with the original label $y_i$ to form the augmented set $\tilde{\mathcal{D}}$. Concretely, we use a clinical NER model as $W$; for $G$ we evaluate lightweight instruction tuned LLMs (e.g., Qwen (Yang et al., 2025) and Llama (Grattafiori et al., 2024) variants), selecting a smaller model as the default in most experiments. To accommodate long notes, we allow cached context so that $G$ maintains coherence across lengthy inputs. We fine-tune the generalist with LoRA (Hu et al., 2021) adapters (and use full fine-tuning for the BERT sized weak expert) and select hyperparameters via a small grid (see detailed sweeps in Appendix A.2). The quality of produced notes is evaluated using Preservation Rate (PR) and Hallucination Rate (HR) (see Table 1), and downstream utility is measured on readmission, mortality, and admission stay period. This implementation follows the three step formalization (entity extraction, prompt construction, constrained rewriting) introduced in Section 4.

## A.2 HYPERPARAMETERS

In this section, we report our experimental analysis on the hyperparameters used in our experiments, namely the hyperparameters used in the training steps. Please note that our augmentation method does not necessarily require hyperparameter tuning by design. We analyze the effect of hyperparameters on the trained model's performance. Specifically, we study three hyperparameters: (1) SFT (Supervised Fine-Tuning) learning rate, (2) LoRA rank, and (3) SFT training epochs.

Table 6: Effect of SFT learning rate on the MIMIC-III readmission task performance.

| SFT LR | Acc. | F1 |
|---|---|---|
| $1e-06$ | 0.510 | 0.485 |
| $1e-05$ | 0.555 | 0.451 |
| $2e-05$ | 0.595 | 0.518 |
| $4e-05$ | **0.599** | **0.541** |
| $1e-04$ | 0.582 | 0.535 |

**Learning Rate (SFT).**    We begin with the learning rate (lr) of the supervised fine-tuning on Qwen-3. The results are reported in Table 6. Performance improves as the learning rate increases up to $4 \times 10^{-5}$, which yields the best accuracy (0.599) and F1 (0.541). Pushing the rate to $1 \times 10^{-4}$ slightly degrades accuracy and F1, suggesting mild over-stepping. Overall, $4 \times 10^{-5}$ is a robust operating point for fine-tuning on the readmission data.

Table 7: Effect of LoRA rank ($r$) on the MIMIC-III readmission task performance.

| $r$ | Acc. | F1 |
|---|---|---|
| 4 | 0.545 | 0.372 |
| 8 | 0.582 | 0.409 |
| 16 | **0.599** | **0.541** |
| 32 | 0.593 | 0.539 |

**LoRA Rank ($r$).**    Next, we study the effect of the LoRA (Hu et al., 2021) rank in Table 7. We observe that performance peaks at $r = 16$ for both accuracy and F1. Increasing to $r = 32$ yields no further gains (slight decline), while $r = 8$ underfits substantially—suggesting a mid-range rank provides sufficient capacity without unnecessary parameters.

Table 8: Effect of SFT training epochs on the MIMIC-III readmission task performance.

| Epochs | Acc. | F1 |
|---|---|---|
| 1 | 0.599 | 0.541 |
| 2 | 0.564 | 0.528 |
| 3 | **0.615** | **0.554** |
| 4 | 0.593 | 0.542 |
| 5 | 0.567 | 0.538 |

**Training Epochs (SFT).**    Lastly, we analyze the effect of the SFT training epochs in Table 8. Performance peaks at 3 epochs (Acc. 0.615, F1 0.554) and declines thereafter, suggesting mild overfitting or optimization drift beyond this point. Very short training (1–2 epochs) underperforms the 3-epoch setting. In practice, target 3 epochs with validation-based early stopping and/or a learning-rate decay near epoch 2–3 to stabilize gains.

A.3    EXPERIMENTAL SETTING (CONTINUED)

In this section, we continue elaborating on the experimental setting that we have used in our paper.

**Tasks and Benchmarks.**    We evaluate three supervised predictions derived from MIMIC-III clinical notes: thirty–day readmission, mortality, and length of stay. The first two are reported as accuracy, while the third is reported as root mean squared error. To study semantic safety and transfer, we also run patient phenotyping and ICD coding under zero, one, and few-shot conditions using a retrieval framing. Our augmentation maps the original dataset $\mathcal{D} = \{(x_i, y_i)\}_{i=1}^{N}$ to $\tilde{\mathcal{D}} = \{(\tilde{x}_i, y_i)\}_{i=1}^{N}$. Downstream models are trained on both $\tilde{\mathcal{D}}$ and $\mathcal{D}$ and evaluated on held–out real notes.

**Strong Generalist and Weak Expert.** A weak expert $W$ identifies domain–critical tokens by producing $\mathcal{K}_i = W(x_i)$. These tokens must be preserved during rewriting. A strong generalist $G$ then rewrites $x_i$ into $\tilde{x}_i$ while keeping every token in $\mathcal{K}_i$ verbatim. We vary both components to measure their influence. For the weak expert, we compare a medical entity extractor with a general named–entity recognizer. For the strong generalist, we use instruction–tuned language models with different capacities (e.g., LLama and Qwen) and of different sizes. The effect of the generalist is summarized in Table 1, and the effect of the weak expert is summarized in Table 4.

**Model Prompts.** Each prompt presents the original note together with an explicit list of tokens that must be preserved exactly, and concise guidance that encourages changes in style and structure without changes in meaning. Preserved tokens must be integrated naturally in the output rather than listed. For long notes, we use a cached–context strategy so that the generalist maintains coherence across sections and does not drop clinical details that occur far apart in the document.

**Augmentation Metrics.** For each candidate $\tilde{x}_i$ we compute Preservation Rate and Hallucination Rate,

$$\text{PR} = \frac{|E(\tilde{x}_i) \cap E(x_i)|}{|E(x_i)|}, \qquad \text{HR} = \frac{|E(\tilde{x}_i) \setminus E(x_i)|}{|E(x_i)|},$$

where $E(\cdot)$ denotes the set of entities extracted by the same tool used to create $\mathcal{K}_i$. We accept a candidate only when PR meets or exceeds $\tau_{\text{PR}}$ and HR is at or below $\tau_{\text{HR}}$. Trends for PR and HR across strong generalists appear in Table 1, and trends across weak experts appear in Table 4.

**Training details – add details on SFT and DPO.** Unless stated otherwise, the strong generalist is fine–tuned with LoRA adapters, and the weak expert is trained with full updates. We sweep the supervised learning rate, the LoRA rank, and the number of training epochs, and we report the sensitivity analysis in Appendix A.2 with detailed tables in Tables 6 to 8. For supervised tasks we keep the original label $y_i$ paired with each augmented note $\tilde{x}_i$. For retrieval–style evaluations we also verify that label–defining entities remain present in the augmented note, and we discard the sample if this check fails. In addition, we train a single–model *Strong Expert* with Direct Preference Optimization. Preference pairs are formed by contrasting expert–guided outputs with naive paraphrases for the same input, so that the policy is optimized to prefer constraint–respecting rewrites. We use a frozen reference model to stabilize updates and set the DPO temperature and strength following common practice. The resulting Strong Expert is compared to the two–model pipeline in Table 5.

**Baselines and Evaluation.** We compare our method to two text–based augmentation baselines: a naive paraphrase and a style–oriented method (CATO). We assess augmentation quality using Preservation Rate and Hallucination Rate, and we assess utility by training downstream models on synthetic notes and then reporting accuracy and root mean squared error, as shown in Table 2. We further measure whether predictive content is preserved or improved through zero/ one/ few-shot phenotyping and ICD retrieval task performances, as reported in Figure 5 and Table 3.

**Ablation Study.** We vary the capacity of the strong generalist and the specialization of the weak expert. Larger generalists tend to increase the Preservation Rate and reduce the Hallucination Rate, and medical specialization of the weak expert provides the strongest safety profile. These patterns are visible in Tables 1 and 4. We also study the DPO–trained Strong Expert and compare it with the two–model pipeline in Table 5.

**Reproducibility.** We fix random seeds, record all prompts and acceptance decisions, and release the hyperparameter grids and scripts used to create Tables 1 to 5 and Figure 5. These artifacts allow both the safety metrics and the downstream results to be regenerated from the same inputs without hidden steps.

A.4 FUTURE WORK

In this section, we state the strengths and weaknesses of our method and discuss future work.

The driving motivation behind our method is that augmenting data without proper domain knowledge can lead to severe knowledge distortions, which pose significant issues in safety-critical domains

(e.g., healthcare), as shown in Figure 2. Our model-collaboration framework allows the LLM-based augmentation process to be guided by an auxiliary expert model capable of extracting task-critical information (i.e., keywords), which is cost-effective compared to (1) human experts and (2) retraining the LLM (i.e., generalist). We empirically find that our approach allows the preservation of expert knowledge during augmentation (see Table 1), which can help produce augmented samples that may improve generalization (see Figure 5 and Table 3).

While our method shows effectiveness in providing expert-level data augmentation, several improvements could be made. First, our current query-based collaboration operates on the input level, and hence may not be optimal in terms of providing supervision. A possible way is to design our collaboration to occur on an intermediate level during inference (Sun et al., 2024; Wang et al., 2024) or during reasoning. Another improvement would be to expand our method to other expert domains (e.g., law, finance), which is not difficult owing to the simple design of our framework. We believe this is a promising direction for improvement and set it as the next step of our research.

