# OpenReview forum: "Expert-guided Clinical Text Augmentation via Query-Based Model Collaboration"
_ICLR.cc/2026/Conference — Submitted to ICLR 2026_

### Official Review · Reviewer_GaEP · 2025-10-28

**Soundness:** 3
**Presentation:** 4
**Contribution:** 3
**Rating:** 4
**Confidence:** 4

**Summary:**

This work addresses the challenge of safe data augmentation (avoiding introduction of factual errors and hallucinations) for clinical text using large language models. The idea is existing LLM-based augmentation methods lack domain-specific safeguards and thus may delete or introduce medical entities that should not be present in the clinical notes. To address this problem, the authors propose a collaboration framework that combines a weak expert to identify important medical entities in the original clinical note and a general-purpose large language model to rewrite the clinical note while being explicitly asked to preserve the medical entities identified by the weak expert. The data augmentation is evaluated on MIMIC-III clinical notes across several different downstream tasks. The paper also demonstrates the collaboration framework can be distilled into a single model via Direct Preference Optimization, although it underperforms the 2-model collaboration framework.

**Strengths:**

* The problem is well motivated as the safety risks of hallucinations and medical term deletions in clinical text augmentation has not been studied previously.
* The framework is practical as it uses a lightweight named entity recognition model to guide a strong generalist model, which does not require the costly annotation compared to human expert supervision or full LLM training (or re-training).
* Reasonable experimental validation setup with demonstration across three prediction tasks, different model architectures with zero-shot and few-shot scenarios.
* Clear presentation as the paper is relatively well-structured and has effective use of figures to convey the message.

**Weaknesses:**

* The motivation for weak expert versus a retrieval-augmented generation approach to inject domain knowledge isn't quite clear. In addition, is there a reason why generalist model is preferred and not one of the medical-domain focused large language model? There has also been work that incorporates medical graph knowledge to perform data augmentation as well (see Varshney et al. in Scientific Reports 2023, Yu et al. in ACM Computing Surveys 2022, and Xu et al. in KDD 2025) but it's not clear why the NER weak expert might be preferable to these other two other approaches.
* The evaluation metrics have some limitation as it relies purely on the NER model to identify whether it was extracted from the original text, and clearly providing a prompt with these entities will improve preservation rate and hallucination rate. Is there any human evaluation of clinical correctness, or potentially using multiple LLM as judges to identify that these are in fact better data augmentation?
* The baselines are a bit weak in that it compares against simple rephrasing and style-based augmentation. There are other variants that use RAG approaches or graph approaches to achieve data augmentation. Is this approach comparable or better than those? Similarly, do you need the weak expert if you have a medical-domain LLM model? It's also not clear why larger LLMs aren't tried if the goal is to perform data augmentation without requiring any fine-tuning.
* Only one NER backbone is chosen as the weak expert (another uses just a simple language mode). An ablation on the effect of the weak entity extraction would be useful to better understand the performance characteristics of the model as there have been various biomedical NER models like Bioalbert, BioBert, MedBert.
* Causal graph is mentioned but is not deeply integrated into the methodology nor the experiments. What is the function of the causal graph in the context of data augmentation?
* The experimental setup isn't quite clear. How many data augmentations are constructed and passed to the model? For the downstream tasks, is it few shot demonstrations? Is it a combination of augmented and real data?

**Questions:**

1. Why did you choose a weak expert NER approach over (i) retrieval-augmented generation (RAG) for injecting domain knowledge; (ii) a medical-domain LLM (or biomedical-domain LLM), and (iii) incorporating medical knowledge graphs for data augmentation?
2. In your framework, does having larger, more capable LLMs that might have better inherent medical knowledge perform just as well as your NER-based version?
3. The preservation rate and hallucination rate metric rely on the same NER model used for extraction. Does the results still hold when either using different NER Models for extraction or when evaluated by human clinicians?
4. Does your framework generalize to various NER backbones? As suggested in Table 4, the effects is minimal when using a general expert which suggests there is some necessary bare minimum as it barely outperforms the Naive version.
5. How is the causal graph related to the methodology? Are you positing that all entities are considered causal?
6. What is the experimental setup in terms of how much data is augmented / created and how are the prediction tasks measured?

---

> ### Author Response · Authors · 2025-11-20
> **[for Reviewer GaEP] Response to Weaknesses 1-2**
>
> ## Weaknesses
> ***
> ### 1. Weak Motivation: why using a weak expert? / Baselines
> > The motivation for weak expert versus a retrieval-augmented generation approach to inject domain knowledge isn't quite clear...
>
> > In your framework, does having larger, more capable LLMs that might have better inherent medical knowledge perform just as well as your NER-based version?
>
> > do you need the weak expert if you have a medical-domain LLM model?
>
> Thank you for your suggestion. Reflecting your concern, we have added two baselines to show that using a model-collaborative framework performs better than retrieval-augmented generation (RAG) for injecting domain knowledge [1] and a medical-domain LLM (BioMistral-7B-DARE). As shown in Table 1, our method remains robust in preserving critical medical concepts, and Table 2 shows that data augmented with our method yields the largest downstream performance gains. We will update the experimental sections in the paper.
>
> An interesting observation is that though BioMistral-7B-DARE is larger than our augmentation model (LLaMA-3B) and is trained on biomedical corpora, it exhibits a high hallucination rate and alters critical clinical information (Table 1). This is likely because larger biomedical LLMs are optimized for fluent generation and may paraphrase, elaborate, or fill in missing details, which increases the chance of altering clinical facts when no explicit constraints are provided.
>
> The SynName+SciName method based on RAG-retrieved synonym substitution [1] is much safer (low hallucination in Table 1) but offers only limited downstream improvement (Table 2), suggesting that word-level substitution alone is insufficient to meaningfully enrich data or improve robustness to real clinical note variation.
>
> Overall, the added baselines clarify our contribution: we achieve both token-level safety and document-level augmentation capability, addressing limitations of existing approaches.
>
> <small>Table 1: Quality of synthetic notes generated by different augmentation methods, measured by medical concept preservation rate (PR) and
> hallucination rate (HR) across 300 samples.</small>
> | Method | PR ↑ | HR ↓ |
> | -------- | -------- | -------- |
> | Naive     | 0.559 | 0.290  |
> | CATO     |  0.568  | 0.383  |
> | Biomedical LLM   |  0.425   |   0.367   |
> | SynName+SciName [1]     |   0.667   |   0.275   |
> | Ours (medical expert)  |  **0.725**  |  **0.263**  |
>
>
> *Table 2: Downstream task performance (Acc./RMSE) of Qwen-3 model trained with augmented data using different methods.*
>
> | Aug. Method | Readmission (Acc.) | Mortality (Acc.) | Length-of-stay (RMSE) |
> | -------- | -------- | -------- | -------- |
> | Zero-shot  | 0.511 | 0.901 | 73.277 |
> | None     | 0.526 | 0.911 | 17.835 |
> | Naive     | 0.520 | 0.907 | 16.357 |
> | CATO     |  0.552  | 0.910 | 18.677 |
> | Biomedical LLM   |  0.538  | 0.914 | 19.324 |
> | SynName+SciName [1]    | 0.527   | 0.905 | 17.990  |
> | Ours  |  **0.599**  | **0.917** | **15.563**  |
>
>
> [1] Abdollahi, Mahdi, et al. "Substituting clinical features using synthetic medical phrases: Medical text data augmentation techniques." Artificial Intelligence in Medicine 120 (2021): 102167.
>
>
> ***
>
> ### 2. Qualitative Metric via Human Evaluation
> > The evaluation metrics have some limitation as it relies purely on the NER model to identify ...
>
> Thank you for raising the important point! We agree that human evaluation is valuable for assessing clinical safety and we have started clinician annotation loop. We will update the paper with the results as soon as they are available.

---

> ### Author Response · Authors · 2025-11-20
> **[for Reviewer GaEP] Response to Weaknesses 3-5**
>
> ### 3. Ablation on Weak Expert
> > Only one NER backbone is chosen as the weak expert (another uses just a simple language mode). An ablation on the effect of the weak entity extraction would be useful...
>
> Thank you for your suggestion. In Table 4, we have reported an ablation study of how the variation in weak expert affects our method's effectiveness.
>
> *Table 4: Effects of using different weak-expert models (biomedical-ner-all, Medical-NER, and BioMed-NER). Weak expert proficiency in extracting expert knowledge affects the quality of data augmentation.*
> | Method              | PR ↑ | HR ↓ |
> |---------------------|:----:|:----:|
> | Naive               | 0.51 | 0.59 |
> | CATO                | 0.62 | 0.77 |
> | Ours (general-expert) | 0.53 | 0.50 |
> | Ours (biomedical-ner-all) | 0.79 | 0.33 |
> | Ours (Medical-NER)   | 0.68 | 0.39
> | Ours (BioMed-NER)   | 0.77 | 0.42
>
> The results indicate that different clinical NER models provide similar inductive signals. When used as medical weak experts, all three NERs (biomedical-ner-all, Medical-NER, BioMed-NER) improve PR and reduce HR relative to Naive and CATO, indicating that the performance gain is not tied to a specific NER family.
>
> ***
> ### 4. Causal Graph
> > Causal graph is mentioned but is not deeply integrated into the methodology nor the experiments. What is the function of the causal graph in the context of data augmentation?
>
> Thank you for the question. The causal graph in Figure 1 represents the problem setup and pipeline diagram, focusing on the safety NLP issue we address. The methodology involves altering only non-causal text while protecting clinically critical spans, as defined by different variables in the causal graph. We will clarify in Section 3 how readers should use the causal graph to understand the setup.
>
> ***
> ### 5. Experimental Setup
> >The experimental setup isn't quite clear. How many data augmentations are constructed and passed to the model? For the downstream tasks, is it few shot demonstrations? Is it a combination of augmented and real data?
>
> We generated 10,000 augmented samples and use them along with the original samples to train models for each prediction task. Then, we assess the effect of augmented data by evaluating the trained model on the downstream tasks (see Tab. 2). We also tested the quality of augmented data using zero/few-shot inference (see Tab.3). We will update the paper to better clarify our experimental setups.

---

> ### Author Response · Authors · 2025-11-20
> **[for Reviewer GaEP] Response to Questions 1-4**
>
> ## Questions
> ***
> ### 1. Why did you choose a weak expert NER approach over (i) retrieval-augmented generation (RAG) for injecting domain knowledge; (ii) a medical-domain LLM (or biomedical-domain LLM), and (iii) incorporating medical knowledge graphs for data augmentation?
>
> Thank you for your question. Please see our response to weakness point 1.
>
> ***
> ### 2. In your framework, does having larger, more capable LLMs that might have better inherent medical knowledge perform just as well as your NER-based version?
>
> As in our response to weakness point 1, using a  larger, more capable LLMs that might have better inherent medical knowledge does not gurantee that it will produce safe and semantic-preserving augmentation.
>
> ***
> ### 3. The preservation rate and hallucination rate metric rely on the same NER model used for extraction. Does the results still hold when either using different NER Models for extraction or when evaluated by human clinicians?
>
> Thank you for your suggestion, we initially use biomedical-ner-all for computing the preservation rate and hallucination rate metrics, and we agree that an ablation regarding the HR/PR estimator (NER model) would help show the robustness of our method. The ablation result is in Table 6:
>
> *Table 6: Effects of using a different weak-expert model (BioMed-NER) **for HR/PR estimation*** on the augmented data.
> | Method            | PR ↑ | HR ↓ |
> |---------------------|:----:|:----:|
> | Naive               | 0.43 | 0.55 |
> | CATO                | 0.59 | 0.73 |
> | Ours (general-expert) | 0.51 | 0.48 |
> | Ours (biomedical-ner-all) | 0.76 | 0.40 |
> | Ours (Medical-NER)   | 0.70 | 0.42
> | Ours (BioMed-NER)   | 0.72 | 0.46
>
> As expected, the absolute PR/HR values differ when evaluated with a different NER model, because each NER defines the set of clinical entities slightly differently, **but the relative trends are very stable**. In the table, all variants of our method consistently achieve higher preservation rates and lower hallucination rates than Naive and CATO.  This indicates that HR/PR capture genuine differences in hallucination behavior rather than artifacts of a particular NER model, and that our method remains the best in terms of both preserving clinically relevant entities (PR) and reducing hallucinations (HR) even when the evaluator is changed.
>
> Furthermore, in your next question, we provide additional results on using different weak-expert models **for augmentation**.
>
> ***
> ### 4. Does your framework generalize to various NER backbones? As suggested in Table 4, the effects is minimal when using a general expert which suggests there is some necessary bare minimum as it barely outperforms the Naive version.
>
> Our method is generalizable to various biomedical NER models. In Table 4 we perform ablation on different biomedical weak experts used in our method. It shows that to use a domain-specific NER is cruicial, which can distinguish general and biomedical entities thus constrict large generalist during its generation. A general-expert might identify entity boundaries to minimal gain, but is far less effective than a medical expert. Section 6 in the paper also provides a relevant discussion.
>
> *Table 4: Effects of using different weak-expert models (biomedical-ner-all, Medical-NER, and BioMed-NER) for augmentation. Weak expert proficiency in extracting expert knowledge affects the quality of data augmentation.*
> | Method              | PR ↑ | HR ↓ |
> |---------------------|:----:|:----:|
> | Naive               | 0.51 | 0.59 |
> | CATO                | 0.62 | 0.77 |
> | Ours (general-expert) | 0.53 | 0.50 |
> | Ours (biomedical-ner-all) | 0.79 | 0.33 |
> | Ours (Medical-NER)   | 0.68 | 0.39
> | Ours (BioMed-NER)   | 0.77 | 0.42

---

> ### Author Response · Authors · 2025-11-20
> **[for Reviewer GaEP] Response to Questions 5-6**
>
> ### 5. How is the causal graph related to the methodology? Are you positing that all entities are considered causal?
>
> The causal graph illustrates the core idea behind our methodology: we separate causal features (information relevant to the clinical prediction task) from non-causal features (stylistic and formatting variations). Our method modifies only the non-causal components to increase data diversity and improve robustness to domain shifts (e.g., differences in documentation style across hospitals). At the same time, we explicitly preserve the causal features, operationalized as biomedical entities, to ensure the augmented data remains clinically safe. We are not asserting that all entities are causal in a strict clinical sense, but that they serve as a practical and clinically meaningful approximation for task-relevant content in our implementation.
>
> ***
> ### 6. What is the experimental setup in terms of how much data is augmented / created and how are the prediction tasks measured?
>
> We generated 10,000 augmented samples and combine them with the original training set to fine-tune BERT and Qwen for each prediction task. For mortality prediction, we remove the DEATH indicator from the notes to avoid label leakage; for length-of-stay prediction, we remove ADMITTIME and DISCHTIME. We reserve 10% of the dataset as a separate test set for evaluation. We will add these experimental details to Appendix A.3 (Experimental Setting).

---

### Official Review · Reviewer_rxnD · 2025-10-30

**Soundness:** 1
**Presentation:** 3
**Contribution:** 2
**Rating:** 2
**Confidence:** 4

**Summary:**

This paper proposes an expert-guided framework for clinical text augmentation. A biomedical NER model marks critical medical entities, and an LLM rephrases the note without altering them. The generated notes are used to train downstream predictors on MIMIC-III (e.g., readmission, mortality, length-of-stay), showing modest performance gains and reduced hallucination compared with Naive and CATO baselines. The authors also explore distilling the two-model process into a single model via preference-based reinforcement learning.

**Strengths:**

s1. The paper’s starting point is to ensure the safety and factual correctness of LLM-based augmentation in clinical context, which identifies a genuine challenge in using LLMs for high-stakes domain like healthcare.

s2. The overall presentation is clear: the problem setup, pipeline diagram (Fig. 1), and qualitative examples (Fig. 2) effectively convey the workflow and intended behavior.

s3. The downstream experimental setup is comprehensive, covering multiple clinical prediction tasks.

**Weaknesses:**

W1. The motivation for augmentation lacks sufficient justification for a high-stakes clinical domain. While benefits like robustness are understood in data augmentation studies, practitioners will always prioritize the safety of the original text over the modest performance gains (Table 2) from a synthetic dataset. This is especially true given the risk of factual alterations (see W2). The paper must provide a stronger, necessary motivation for modifying notes, such as supporting de-identification for data sharing or ML model development.

W2. There is inconsistent evidence regarding the paper’s safety claims.

> Appendix A.1: Do not change any medical terminology, dosages, measurements, or clinical findings.

> Figure 2: “fever → pyrexia” and “shortness of breath → dyspnea”

Such replacements violate the stated constraint and introduce risks that undermine the safety argument.

W3. Given replacements in W2, a human evaluation is necessary (at least on a small scale). Safety promised by PR/HR lacks real-world credibility, and healthcare applications just tend to favor correctness over marginal performance gains.

W4. The methodological novelty is incremental. The proposed pipeline essentially combines a general-purpose LLM with an external NER model acting as a token-level constraint.

**Questions:**

See weaknesses.

---

> ### Author Response · Authors · 2025-11-20
> **[for Reviewer rxnD] Response to Weaknesses 1-2**
>
> ## Weaknesses
>
> ***
> ### 1. Lack of Motivation
> > "The motivation for augmentation lacks sufficient justification for a high-stakes clinical domain.."
>
> We agree that safety and factual fidelity are essential in clinical settings. Our motivation for augmentation is not to replace original notes, but to enable improved model development including under realistic data-access constraints and limitations. Beyond the examples cited in the paper, prior clinical and biomedical work has shown that well-controlled synthetic clinical text can preserve task utility while reducing privacy risks and supporting responsible data sharing [1]. For instance, [2] demonstrate that synthetic augmentation can maintain clinical utility while lowering re-identification risks, and literature reviews also highlight its value for addressing data scarcity and data-sharing restrictions in healthcare AI [3].
>
> Importantly, national health agencies have begun adopting synthetic data for research access and privacy protection. U.S. federal organizations such as CDC-NCHS and AHRQ have released public-use synthetic health datasets to enable broader access while reducing disclosure risk, and NHS England is piloting synthetic data for analysis and clinical research (https://digital.nhs.uk/services/artificial-data). These precedents demonstrate that synthetic data is under active consideration and development within clinical and public-health institutions.
>
> We will revise the motivation section to incorporate this literature and clarify the role of synthetic augmentation in safe clinical AI development.
>
>
> ### [References]
> [1] Qian, Z. et al. (2024). Synthetic data for privacy-preserving clinical risk prediction. Scientific Reports, 14, 25676.
>
> [2] Kim, W., Hahm, S., & Lee, J. (2024). Generalizing clinical de-identification models by privacy-safe data augmentation using GPT-4. EMNLP 2024.
>
> [3] Pezoulas, V. C. et al. (2024). Synthetic data generation methods in healthcare: A review on open-source tools and methods. Computational and Structural Biotechnology Journal, 23, 2892–2910.
>
> [4] Nisevic, M., Milojevic, D., & Spajic, D. (2025). Synthetic data in medicine: Legal and ethical considerations for patient profiling. Computational and Structural Biotechnology Journal.
>
> [5] Giuffrè, M., & Shung, D. L. (2023). Harnessing the power of synthetic data in healthcare: innovation, application, and privacy. NPJ Digital Medicine, 6, 186.
>
> ***
> ### 2. Inconsistent evidence regarding the paper’s safety claims
> > "Appendix A.1: Do not change any medical terminology, dosages, measurements, or clinical findings."
> > "Figure 2: “fever → pyrexia” and “shortness of breath → dyspnea”
>
> Thank you for the question. The prompt instruction in Appendix A.1 (“Do not change any medical terminology…”) is intended to prevent the model from altering clinical meaning, such as modifying dosages or introducing new medical findings. However, in practice, large language models may still produce **clinically equivalent synonyms** (e.g., fever → pyrexia, shortness of breath → dyspnea).
>
> This behavior does not violate our safety criterion, because our evaluation is performed at the **UMLS concept level**. Synonyms that map to the same medical concept preserve the underlying clinical information and are therefore considered acceptable.
>
> To avoid confusion, we will revise the appendix to clarify that the constraint aims to prohibit semantic changes (e.g., different diagnoses, different measurements), while terminology-level paraphrases that preserve UMLS concepts are allowed. As shown in Table 1, our method achieves high concept-preservation scores, confirming that these substitutions do not affect clinical meaning.
>
> *Table 1: Quality of synthetic notes generated by different augmentation methods, measured by medical concept preservation rate (PR) and
> hallucination rate (HR) across 300 samples.*
> | Method | PR ↑ | HR ↓ |
> | -------- | -------- | -------- |
> | Naive     | 0.559 | 0.290  |
> | CATO     |  0.568  | 0.383  |
> | Biomedical LLM   |  0.425   |   0.367   |
> | SynName+SciName     |   0.667   |   0.275   |
> | Ours (medical expert)  |  **0.725**  |  **0.263**  |

---

> > ### Comment · Reviewer_rxnD · 2025-11-27
> >
> > I thank the authors for the responses, especially #2 regarding safety. Happy to see my other comments partially addressed as well, and happy to raise the overall evaluation.

---

> ### Author Response · Authors · 2025-11-20
> **[for Reviewer rxnD] Response to Weaknesses 3-4**
>
> ### 3. Human Evaluation
> >"Given replacements in W2, a human evaluation is necessary (at least on a small scale). Safety promised by PR/HR lacks real-world credibility, and healthcare applications just tend to favor correctness over marginal performance gains."
>
> Thank you for the suggestion. We agree that human evaluation is valuable for assessing clinical safety. We are currently conducting a small-scale evaluation with our collaborating clinicians and will update the paper with the results as soon as they are available.
>
> ***
> ### 4. methodological novelty is incremental
> > "The proposed pipeline essentially combines a general-purpose LLM with an external NER model acting as a token-level constraint."
>
>
> While many general augmentation techniques exist, many are LLM-based methods are saturated, showing very limited difference between methods [6] in terms of improving downstream task performance and augmentation quality. Furthermore, documentation-level augmentation methods are not commonly used in safety-critical settings such as healthcare due to its innate safety-related issues (e.g., hallucination, factual distortion [7]) and limited effect [8]. This highlights a gap: methods that can both produce meaningful document-level diversity and offer token-level safety guarantees. Our contribution addresses precisely this gap. This combination of token-level safety constraints with document-level expressivity goes beyond simply pairing an LLM with an NER model and is non-trivial in the clinical context, where small factual shifts can compromise safety.
>
> ### [References]
> [6] Ding et al., Data Augmentation using LLMs: Data Perspectives, Learning Paradigms and Challenges, ACL (2024).
>
> [7] Ehrett et al., Leveraging Open-Source Large Language Models for Data Augmentation in Hospital Staff Surveys: Mixed Methods Study, JMIR (2024).
>
> [8] Feder et al., Data Augmentations for Improved (Large) Language Model Generalization, NeurIPS (2023).

---

### Official Review · Reviewer_iHnY · 2025-11-02

**Soundness:** 3
**Presentation:** 3
**Contribution:** 3
**Rating:** 6
**Confidence:** 2

**Summary:**

The paper proposes a query-based model collaboration framework for safe clinical text augmentation. A weak expert (clinical NER on DistilBERT) extracts medical entities to preserve verbatim; a strong generalist (small instruction-tuned LLM, e.g., Qwen-3-0.6B / Llama-3.2-3B) rewrites the note while obeying these constraints. The authors evaluate on MIMIC-III for readmission, mortality, and length-of-stay, plus zero/one/few-shot phenotyping and ICD coding. They report higher entity preservation (PR), lower hallucination rate (HR), and better downstream performance than a naive paraphrase and CATO-style style-only augmentation. They also distill the collaboration into a single model via DPO, with smaller but still positive gains.

**Strengths:**

Clear problem framing for safety-critical NLP: focuses augmentation on non-causal text while protecting clinically critical spans.

Simple, practical pipeline: off-the-shelf clinical NER + small LLM; no heavy training needed to get benefits.

Safety-oriented metrics (PR/HR) and qualitative examples convincingly show typical LLM paraphrase failures and how constraints help.

**Weaknesses:**

Self-evaluation bias in PR/HR: the same NER family is used to (a) define constraints and (b) score preservation/hallucination → optimistic estimates; misses concept-level paraphrases and dosage/unit variants.

Over-constrained “verbatim” preservation: requiring exact tokens may block legitimate clinical synonymy (e.g., myocardial infarction vs MI), risking brittle rewrites and hidden meaning drift.

Baselines are narrow: lacks strong constraint-aware or copy-preserving paraphrase baselines (span locking/bracketing, constrained decoding, mask-and-infill with protected spans, entailment-constrained rewrite, rule/UMLS-guided paraphrase).

**Questions:**

NER–metric coupling: How do PR/HR change if you (i) evaluate with a different clinical NER, (ii) map to UMLS concepts (string-independent), or (iii) use entailment-based factuality checks?

Synonymy & concept-level preservation: Can you preserve concepts (UMLS/RxNorm/SNOMED IDs) rather than exact tokens to allow safe synonym substitutions while protecting meaning?

Stronger baselines: Please add (and report) results for recent papers for a fair comparsion

---

> ### Author Response · Authors · 2025-11-20
> **[for Reviewer iHnY] Response to Weaknesses 1 and 2**
>
> ## Weaknesses and Questions
>
> ***
> ### 1. Evaluation of PR/HR based on concept preservation
> > "misses concept-level paraphrases and dosage/unit variants."
>
> > "Can you preserve concepts (UMLS/RxNorm/SNOMED IDs) rather than exact tokens to allow safe synonym substitutions..?"
>
> Thank you for the insightful suggestions. To address the concern of “over-constrained verbatim preservation,” which may penalize legitimate clinical synonymy, we revised our hallucination and preservation metrics by mapping critical keywords to UMLS concepts. This allows valid synonym substitutions as long as the underlying medical concept is preserved. For entities such as lab values (e.g., *26.3 g/dL*) that lack UMLS concepts, we still measure exact token preservation. We have updated the results in Table 1 accordingly, which show that our method remains robust in preserving critical medical concepts.
>
> *Table 1: Quality of synthetic notes generated by different augmentation methods, measured by medical concept preservation rate (PR) and hallucination rate (HR) across 300 samples.*
> | Method | PR ↑ | HR ↓ |
> | - | - | - |
> | Naive     | 0.559 | 0.290  |
> | CATO     |  0.568  | 0.383  |
> | Biomedical LLM   |  0.425   |   0.367   |
> | SynName+SciName     |   0.667   |   0.275   |
> | Ours (medical expert)  |  **0.725**  |  **0.263**  |
>
> For the inquiry of whether we can preserve concepts rather than exact tokens for augmentation, we agree this is a meaningful alternative direction. Achieving concept-level preservation during generation would require a substantially different architecture, such as integrating a retrieval or concept-linking module, which is beyond the scope of our model-to-model query method. Our current contribution focuses on stable, token-level controllability, and we already incorporate UMLS-based concept mapping in the evaluation to allow legitimate synonym substitutions. Therefore, we view concept-level preservation at generation time as a promising avenue for future work.
>
>
>
> ***
> ### 2. Narrow Baselines
> > "Baselines are narrow: lacks strong constraint-aware or copy-preserving paraphrase baselines (span locking/bracketing, constrained decoding, mask-and-infill with protected spans, entailment-constrained rewrite, rule/UMLS-guided paraphrase)."
>
> Thank you for your suggestion. We agree that adding several representative baselines would strengthen our work. In response, we have conducted additional experiments with (1) RAG-based knowledge-injection augmentation for token-level substitutions [1], and (2) augmentation using the large biomedical LLM BioMistral-7B-DARE without weak expert guidance. Table 1 shows that our method remains robust in preserving critical medical concepts, and Table 2 shows that data augmented with our method yields the largest downstream performance gains. We will update the experimental sections in the paper.
>
> An interesting observation is that though BioMistral-7B-DARE is larger than our augmentation model (LLaMA-3B) and is trained on biomedical corpora, it exhibits a high hallucination rate and alters critical clinical information (Table 1). This is likely because larger biomedical LLMs are optimized for fluent generation and may paraphrase, elaborate, or fill in missing details, which increases the chance of altering clinical facts when no explicit constraints are provided.
>
> The SynName+SciName method based on synonym substitution baseline is much safer (low hallucination in Table 1) but offers only limited downstream improvement (Table 2), suggesting that word-level substitution alone is insufficient to meaningfully enrich data or improve robustness to real clinical note variation.
>
> Overall, the added baselines clarify our contribution: we achieve both token-level safety and document-level augmentation capability, addressing limitations of existing approaches.
>
> *Table 1: Quality of synthetic notes generated by different augmentation methods, measured by medical concept preservation rate (PR) and hallucination rate (HR) across 300 samples.*
> | Method | PR ↑ | HR ↓ |
> | - | -| - |
> | Naive     | 0.559 | 0.290  |
> | CATO     |  0.568  | 0.383  |
> | Biomedical LLM   |  0.425   |   0.367   |
> | SynName+SciName [1]    |   0.667   |   0.275   |
> | Ours (medical expert)  |  **0.725**  |  **0.263**  |
>
> *Table 2: Downstream task performance (Acc./RMSE) of Qwen-3 model trained with augmented data using different methods.*
>
> | Aug. Method | Readmission (Acc.) | Mortality (Acc.) | Length-of-stay (RMSE) |
> | - | - | - | - |
> | Zero-shot  | 0.511 | 0.901 | 73.277 |
> | None     | 0.526 | 0.911 | 17.835 |
> | Naive     | 0.520 | 0.907 | 16.357 |
> | CATO     |  0.552  | 0.910 | 18.677 |
> | Biomedical LLM   |  0.538  | 0.914 | 19.324 |
> | SynName+SciName [1]    | 0.527   | 0.905 | 17.990  |
> | Ours  |  **0.599**  | **0.917** | **15.563**  |
>
> [1] Abdollahi, Mahdi, et al. "Substituting clinical features using synthetic medical phrases: Medical text data augmentation techniques." Artificial Intelligence in Medicine 120 (2021): 102167.

---

> ### Author Response · Authors · 2025-11-20
> **[for Reviewer iHnY] Response to Weakness 3**
>
> ### 3. NER–metric coupling
> >"Self-evaluation bias in PR/HR: the same NER family is used .."
>
> >"NER–metric coupling: How do PR/HR change if you evaluate with a different clinical NER?"
>
> Thank you for this insightful question. We agree that using a different NER model as the evaluator will naturally change the absolute PR/HR values, because each NER model defines its own notion of clinically critical information. This behavior is expected. Importantly, the evaluation remains fair because all augmentation methods are compared under the same evaluator, so relative differences and trends are preserved.
>
> To further examine the reviewer’s concern, we performed additional experiments using different NER models for generating constraints during augmentation, while evaluating everything using biomedical-ner-all (Table 4). Two key findings emerge:
>
> 1. **Different clinical NER models provide similar inductive signals.**
> When used as weak experts, all three NERs (biomedical-ner-all, Medical-NER, BioMed-NER) improve PR and reduce HR relative to Naive and CATO, indicating that the mitigation effect is not tied to a specific NER family.
>
> 2. **Cross-checking removes the self-evaluation bias concern.**
> When augmentation uses one NER (e.g., BioMed-NER) and evaluation uses another (biomedical-ner-all), our method still substantially outperforms baselines.
> This shows that the gains are not an artifact of “self-scoring,” but reflect genuinely better preservation of medically relevant information.
>
>
> *Table 4: Effects of using different weak-expert models (biomedical-ner-all, Medical-NER, and BioMed-NER) for data augmentation. Weak expert proficiency in extracting expert knowledge affects the quality of data augmentation.*
> | Method              | PR ↑ | HR ↓ |
> |---------------------|:----:|:----:|
> | Naive               | 0.51 | 0.59 |
> | CATO                | 0.62 | 0.77 |
> | Ours (general-expert) | 0.53 | 0.50 |
> | Ours (biomedical-ner-all) | 0.79 | 0.33 |
> | Ours (Medical-NER)   | 0.68 | 0.39
> | Ours (BioMed-NER)   | 0.77 | 0.42

---

### Official Review · Reviewer_d8JU · 2025-11-02

**Soundness:** 3
**Presentation:** 3
**Contribution:** 2
**Rating:** 4
**Confidence:** 4

**Summary:**

The paper tackles the problem of using LLMs for clinical text augmentation, noting that naive paraphrasing or style-focused methods can be dangerous in healthcare settings. The authors propose a two-model pipeline: (1) a lightweight “weak expert” extracts domain-critical tokens from the original note; (2) a “strong generalist” LLM rewrites the note but is constrained through a prompt to keep those tokens intact. They evaluate on MIMIC-III across three prediction tasks and two low-resource settings. The method improves entity preservation and reduces hallucination compared to naive LLM augmentation and CATO, and the augmented data yields better downstream accuracy.

**Strengths:**

1. The studied challenge of clinically safe text augmentation is important; using a lightweight clinical expert to guard a stronger LLM is a natural and well-justified solution.
2. The empirical results on multiple MIMIC-III prediction tasks show consistent gains, indicating that constrained augmentation produces data that actually helps downstream models.
3. The analysis comparing against style-focused augmentation highlights that the method specifically reduces medical entity loss and hallucination, confirming that the design targets the right failure mode.

**Weaknesses:**

1. Some closely related work on clinical text generation and augmentation with LLMs is not discussed. For example, [1] explicitly studies how to inject clinical knowledge before LLM generation, which is conceptually similar to “expert-guided” rewriting. It would strengthen the positioning to contrast their knowledge source and objectives with the proposed expert extraction, and, if feasible, to include it as an additional baseline.
2. Section 3 frames the method with a causal view that separates label-relevant variables (V) from other factors (U), but the implementation approximates V only through biomedical NER. This does not seem like a full realization of the causal intent, because many clinically label-bearing elements, such as severity descriptors and care context, are not always captured as NER spans. The paper would benefit from either broadening the extractor or clarifying that current results are for a narrower, entity-level notion of V.
3. The experimental comparison focuses on naive prompting and CATO-style rewriting, but there are many recent works on LLM-based text data augmentation [2]. Adding a small set of representative recent methods would make the empirical case more robust and show that the gains are not only over older or style-focused augmenters.

[1] Knowledge-Infused Prompting: Assessing and Advancing Clinical Text Data Generation with Large Language Models. ACL Findings 2024.
[2] Text Data Augmentation for Large Language Models: A Comprehensive Survey of Methods, Challenges, and Opportunities. 2025

**Questions:**

See the weaknesses above

---

> ### Author Response · Authors · 2025-11-20
> **[for Reviewer d8JU] Response to Weaknesses and Questions**
>
> ## Weaknesses
>
> ***
> ### 1. Positioning of the paper
> > "Some closely related work on clinical text generation and augmentation with LLMs is not discussed... if feasible, to include it as an additional baseline."
>
> We thank the reviewer for highlighting these related studies. While [1] and our work both explore injecting clinical knowledge into LLM generation, they differ substantially in task scope and data format. The objective of [1] is sentence-level general topic generation (e.g., given an entity “abdominal pain”, LLM produces a sentence “What are the possible causes for abdominal pain?”), but our objective is to rewrite whole patient notes (up to 3000 words in length). On methodology wise, [1] retrieves general clinical topics from external knowledge graphs as its knowledge source, which cannot represent the detailed, patient-specific information required in our task. Incorporating [1] as a baseline would therefore require major architectural changes. We agree with the reviewer that a clearer conceptual contrast strengthens the paper, and will expand the Related Work section accordingly to clarify the different knowledge sources and objectives.
>
> We include an additional baseline [2] which inject clinical knowledge from UMLS dictionary. The results are in the response to point 3.
>
> ### [References]
> [1] Knowledge-Infused Prompting: Assessing and Advancing Clinical Text Data Generation with Large Language Models. ACL Findings 2024.
>
> [2] Abdollahi, Mahdi, et al. "Substituting clinical features using synthetic medical phrases: Medical text data augmentation techniques." Artificial Intelligence in Medicine 120 (2021): 102167.
>
> ***
> ### 2. Clarification regarding the Causal Viewpoint
> > "The paper would benefit from either broadening the extractor or clarifying that current results are for a narrower, entity-level notion of V."
>
> We thank the reviewer for the thoughtful observation. We agree that the causal framing in Section 3 is general, while our implementation operationalizes the variable 𝑉 using biomedical NER. We do not claim to capture all possible information, but incorporating these entities alone already provides a stronger baseline. The NER model we use covers diseases/disorders, signs/symptoms, lab values, severity, clinical events, patient history, etc., which collectively capture most clinically meaningful and label-relevant information. Following the suggestion, we will include the detailed list of entity types in the Methodology section to clearly define 𝑉 in this work.
>
> ***
> ### 3. Additional Baselines
> > "Adding a small set of representative recent methods would make the empirical case more robust and show that the gains are not only over older or style-focused augmenters."
>
> Reflecting the reviewer’s feedback, we added several representative and diverse baselines: a RAG-based token-substitution method [2], rewriting using a large biomedical LLM (BioMistral-7B-DARE) without weak-expert guidance, along with CATO (style augmentation) and a naïve paraphrasing baseline. These baselines cover multiple augmentation paradigms: retrieval-based, large-model rewriting, style-focused, and general paraphrasing. As shown in Tables 1 and 2, our approach produces the highest-quality augmented data and yields the largest downstream performance gains.
>
> More broadly, data augmentation for clinical documentation remains underexplored, and prior work [3] has shown that augmentation can outperform general OOD-robustness methods in healthcare settings. This motivates our focus on developing and benchmarking augmentation methods specifically tailored for clinical notes, an area where strong baselines are still limited.
>
> ### [References]
> [2] Abdollahi, Mahdi, et al. "Substituting clinical features using synthetic medical phrases: Medical text data augmentation techniques." Artificial Intelligence in Medicine 120 (2021): 102167.
> [3] Feder et al., Data Augmentations for Improved (Large) Language Model Generalization, NeurIPS (2023).
>
>
> *Table 1: Quality of synthetic notes generated by different augmentation methods, measured by medical concept preservation rate (PR) and hallucination rate (HR) across 300 samples.*
>
> | Method | PR ↑ | HR ↓ |
> | -------- | -------- | -------- |
> | Naive     | 0.559 | 0.290  |
> | CATO     |  0.568  | 0.383  |
> | Biomedical LLM   |  0.425   |   0.367   |
> | SynName+SciName     |   0.667   |   0.275   |
> | Ours (medical expert)     |  **0.725**  |  **0.263**  |
>
> *Table 2: Downstream task performance (Acc./RMSE) of Qwen-3 model trained with augmented data using different methods.*
> | Aug. Method | Readmission (Acc.) | Mortality (Acc.) | Length-of-stay (RMSE) |
> | -------- | -------- | -------- | -------- |
> | Zero-shot  | 0.511 | 0.901 | 73.277 |
> | None     | 0.526 | 0.911 | 17.835 |
> | Naive     | 0.520 | 0.907 | 16.357 |
> | CATO     |  0.552  | 0.910 | 18.677 |
> | Biomedical LLM   |  0.538  | 0.914 | 19.324 |
> | SynName+SciName    | 0.527   | 0.905 | 17.990  |
> | Ours  |  **0.599**  | **0.917** | **15.563**  |

---

### Author Response · Authors · 2025-11-20
**General Response to all reviewers**

We sincerely thank all reviewers for their constructive feedback. Across reviews, there was a shared remark for our goal of document-level clinical augmentation under strict safety constraints, and for our model collaboration framework, where a weak clinical expert steers a generalist LLM to preserve medically critical content while still producing diverse notes. We appreciate the positive feedback on motivation and methodology contribution, including comments of "*The problem is well motivated* / *identifies a genuine challenge in using LLMs for high-stakes domains like healthcare.*", "*The framework is practical / using a lightweight clinical expert to guard a stronger LLM is a natural and well-justified solution.*"

We also recognize several shared concerns and have revised the paper accordingly:

- **Robustness of Hallucination Rate(HR)/Preservation Rate (PR)**
We now (i) move from token-level to UMLS-based concept PR/HR, allowing safe synonymy, and (ii) add an ablation where PR/HR are recomputed with a different NER evaluator. Absolute scores change, but the relative ranking is stable and our method remains best in both PR and HR.

- **Ablation on Weak Expert**
We add an ablation over different weak experts (biomedical-ner-all, Medical-NER, BioMed-NER, general expert), showing that domain-specific NERs yield the strongest PR/HR, but even a general expert improves over naive rewriting, clarifying generalization across NER backbones.

- **Additional Baselines**
We added additional baselines, including (1) a RAG-based token substitution and (2) domain fine-tuned LLM. Our method uniquely achieves both high concept preservation and the strongest downstream performance. We also expand the motivation section to better connect augmentation to realistic, privacy- and access-constrained clinical workflows.

- **Causal Framework**
We clarify the causal graph by explicitly defining $V$ as the NER-extracted clinically critical entities and explaining that our intervention targets only non-causal text. We detail the augmentation volume and how synthetic and real notes are combined for downstream training. Finally, we have initiated a small-scale clinician evaluation and will include results in the updated version.

We thank everyone for their detailed feedback. We will update the revision, reflecting the shared suggestions of the reviewers. We believe that the strengthened empirical results have addressed reviewers' concerns.

---

### Meta-Review · Area_Chair_PQGN · 2026-01-09

**Summary:**

This work puts forward a  clinical text augmentation approach that first marks  biomedical named entities, and leverages an  LLM to rephrase clinical notes without changing those critical parts of the note.
The work's motivation of ensuring the safety and factual correctness of LLM-based augmentation in clinical context is strong.
Although the paper addresses an important and safety-critical problem, it has  limited conceptual novelty and unresolved concerns around validation and justification. Despite added baselines and rebuttal experiments, the evaluation remains heavily reliant on NER-based automatic metrics. The work has potential, the authors are encouraged to continue their work and resubmit it in the future.

**Reviewer Concerns:**

Authors added new baselines

**Reviewer Scores:**

Unchanged

---

### Decision · Program_Chairs · 2026-01-26

Reject